# Identification of a Novel *Yersinia enterocolitica* Strain from Bats in Association with a Bat Die-Off That Occurred in Georgia (Caucasus)

**DOI:** 10.3390/microorganisms8071000

**Published:** 2020-07-04

**Authors:** Tata Imnadze, Ioseb Natradze, Ekaterine Zhgenti, Lile Malania, Natalia Abazashvili, Ketevan Sidamonidze, Ekaterine Khmaladze, Mariam Zakalashvili, Paata Imnadze, Ryan J. Arner, Vladimir Motin, Michael Kosoy

**Affiliations:** 1Lugar Center for Public Health Research, 0184 Tbilisi, Georgia; tataimnadze@gmail.com (T.I.); ezhgenti@gmail.com (E.Z.); malanial@yahoo.com (L.M.); abazashvilin@yahoo.com (N.A.); ksidamonidze@gmail.com (K.S.); khmaladze.e@gmail.com (E.K.); m.zakalashvili@ncdc.ge (M.Z.); pimnadze@ncdc.ge (P.I.); 2Faculty of Medicine, Public Health and Epidemiology Department, Ivane Javakhishvili Tbilisi State University, 0179 Tbilisi, Georgia; 3Vertebrate Animals Research Group, Institute of Zoology, Ilia State University, 0162 Tbilisi, Georgia; ioseb.natradze@iliauni.edu.ge; 4Ryan Arner Science Consulting LLC, Freeport, PA 16229, USA; ryan.j.arner@gmail.com; 5Department of Pathology, University Texas Medical Branch, Galveston, TX 77555-1019, USA; vlmotin@utmb.edu; 6KB One Health LLC, Fort Collins, CO 80523, USA

**Keywords:** bacterial reservoir, insectivorous bats, *Yersinia entercolitica*, genome sequence, bat die-off, zoonotic agents, Georgia (country)

## Abstract

*Yersinia entercolitica* is a bacterial species within the genus *Yersinia*, mostly known as a human enteric pathogen, but also recognized as a zoonotic agent widespread in domestic pigs. Findings of this bacterium in wild animals are very limited. The current report presents results of the identification of cultures of *Y. entercolitica* from dead bats after a massive bat die-off in a cave in western Georgia. The growth of bacterial colonies morphologically suspected as *Yersinia* was observed from three intestine tissues of 11 bats belonging to the *Miniopterus schreibersii* species. These three isolates were identified as *Y. enterocolitica* based on the API29 assay. No growth of *Brucella* or *Francisella* bacteria was observed from tissues of dead bats. Full genomes (a size between 4.6–4.7 Mbp) of the *Yersinia* strains isolated from bats were analyzed. The phylogenetic sequence analyses of the genomes demonstrated that all strains were nearly identical and formed a distinct cluster with the closest similarity to the environmental isolate O:36/1A. The bat isolates represent low-pathogenicity Biotype 1A strains lacking the genes for the Ail, Yst-a, Ysa, and virulence plasmid pYV, while containing the genes for Inv, YstB, and MyfA. Further characterization of the novel strains cultured from bats can provide a clue for the determination of the pathogenic properties of those strains.

## 1. Introduction

Bats have received intensive attention as reservoirs of zoonotic pathogens, mostly of viral infectious agents. Notoriously, bats are shown to be tolerant to various especially virulent pathogens [1]. At the same time, deadly epidemics among bats are described with massive die-offs from the fungal disease white nose syndrome [2]. A role of bacterial pathogens as sources of fatal infections among bats is less known [3]. Recent reviews listed a number of bacterial agents identified in bats, including those of the genera *Bartonella*, *Borrelia*, *Leptospira*, *Rickettsia*, *Salmonella*, and *Yersinia* [2,4].

*Yersinia entercolitica* is a bacterial species within the genus *Yersinia*, which is mostly known as a human enteric pathogen causing the disease known as “yersiniosis”. In this regard, this bacterium is similar to another representative of this genus, *Y. pseudotuberculosis* that is also known as an enteropathogenic species [5] and very different from the most famous representative of this genus—*Y. pestis*, a pathogen of animals accidently causing cases and occasionally large epidemics among people [5]. As one of the major human foodborne pathogens, *Y. entercolitica* is also recognized as a zoonotic agent widespread in domestic and game animal populations [6]. Specifically, domestic pigs are known as the main reservoir of *Y. entercolitica* [7]. Findings of this bacterium in wild animals are very limited and, to our knowledge, there is only one reported success of culturing *Y. entercolitica* from bat tissues [8]. The current report presents results of identification of bacterial cultures of *Y. entercolitica* from dead bats recovered from a massive die-off in the country of Georgia (Caucasus).

## 2. Materials and Methods

### 2.1. Study Site and Sample Collection

Dead bats were found in a karst cave (Tkibula-Dzevrula) located on the border between Tkibuli and Terjola municipalities of Imereti Region in western Georgia (42.276° N/42.951° E), on the Okriba-Argveta ridge, in the basin of the river Tkibula-Dzevrula (Figure 1 and Figure 2). The cave has three entrances with an interchange of short vertical shafts and inclined steps that are connected to the sub-horizontal gallery. The gallery has many branches that are variable in length and water content. Total length of the cave is 1635 m (5364 ft), with a width of 3–20 m (9.8–65 ft) and a height of 4–15 m (13–49 ft). Stalactites and stalagmites are present. The air is dynamic and the temperature in dry areas varies between 10.4–11.4 °C (50.7–52.52 °F) [9].

At the end of April 2018, we were informed by a local speleologist about a die-off of bats in the cave and 20 dead bodies were collected and kept frozen until transport to the National Center of Diseases Control and Public Health (NCDC) in Tbilisi. The carcasses were morphologically identified by species.

Spleen, liver, and small intestine tissue samples were sterilely collected from the 20 carcasses in a certified Class II Biological safety cabinet at NCDC and processed to special and general microbiology. The samples were processed using aseptic technique to minimize introduction of contaminants into the sample according to the standard operational procedure approved by NCDC.

### 2.2. Bacteriology

For culturing *Francisella tularensis* (the agent of tularemia), homogenized spleen and liver tissues were transferred to cysteine-enriched agar with chocolatized 9% sheep blood. The inoculated agar was incubated at 35 °C for 14 days. To detect *Brucella* species, suspensions from the same tissues were plated on Tryptic Soy Agar, then incubated at 35 °C in 10% CO_2_ for 5 days. Attempts to identify *Brucella* species were justified by a recent report of *Brucella* strain in bats from Georgia [10].

For culturing *Yersinia* species, suspensions of small intestine samples were inoculated to 1% peptone water and incubated at 4 °C for 28 days. Bacterial colonies were sub-cultured on CIN (Cefsulodin, Irgasan, Novobiocin)agar and Endo agar plates at 28 °C on ambient condition for 7 days. Resulting bacterial colonies were Gram-stained and microscopically analyzed. Based on the Gram-staining and bacterial morphology results, Phoenix (Becton, Dickinson and Company Switzerland Sarl, Vaud. Switzerland) and Analytical Profile Index (API; bioMérieux SA, Marcy l’Etoile, France) tests were applied following manufacturers’ instructions for further identification. A bacterial suspension was used to rehydrate each of the wells and the strips were incubated. All positive and negative API test results were compiled to obtain a profile number, which was then compared with profile numbers in a commercial codebook.

For general bacteriology, homogenized tissue suspensions were plated on sheep blood, CIN agar and Endo agar plates and incubated for 72 h at ambient condition following API29 E and Phoenix tests for their further identification.

### 2.3. Genomic DNA Preparation and Sequencing

For whole-genome (WG) sequencing, genomic DNA was extracted using the QIAGEN Genomic DNA Buffer Set and Genomic-tip 100/G, according to manufacturer’s instructions (QIAGEN, Valencia, CA, USA). A 1 µg DNA sample was fragmented using a Covaris ultrasonicator M220 sonication system (Covaris, Inc. Woburn, MA, USA). An Illumina paired-end library was constructed using the NEBNext^®^ Ultra DNA library prep kit for Illumina (New England Biolabs (San Diego, CA, USA). DNA was amplified (5 PCR cycles) using indexed primers, and then purified using an Ampure XP beads (Beckman Coulter). Libraries were quantified using Qubit dsDNA HS kit on a Qubit 3.0 Fluorometer, (Invitrogen, Life Technologies (Malaysia). The library fragment sizes and quality was assessed using a high sensitivity DNA chip on a Agilent 20100 Bioanalyzer (Agilent Technologies, Inc. Germany). Sequencing was performed on Illumina MiSeq platforms, using Miseq reagent kit V2·500 cycles (LC18E NCDC B Int-3; San Diego, California, USA) and V3·600 cycles (samples # LC18E NCDC B Int-1 and LC18E NCDC B Int 9–14). A total of 5,869,674–10,565,252 reads were generated with mean read length of 241.48–277.38 

### 2.4. Plasmid Screening

For screening, the plasmid content of identified *Y. enterocolitica* isolates from the bats, strains were cultured on Tryptic (trypticase) soy agar (TSA) plates and incubated at 28 °C for 24 h. Plasmid DNA isolation was conducted following the protocol of Kado and Liu [11]. *Yersinia pestis* vaccine strain EV 76 was used as a plasmid extraction control and molecular size marker. Electrophoresis of plasmid DNA was performed on a 0.8% agarose gel for 6 h in TBE buffer at 100 V. The gel was stained with ethidium-bromide, visualized under UV trans-illumination and photographed.

### 2.5. Sequence Analysis

Raw sequence reads from Illumina for each sample were processed using the CLC Genomics Workbench 12.0.3 (CLC Bio, Aarhus, Denmark). Low-quality bases were trimmed. High quality reads satisfying the criteria of minimum mean quality score of 30 and minimum read length of 150 bp were retained. Short reads were de novo assembled into contigs (kmer size N 20), and contigs less than 1000 bp were filtered. In total 29, 28 and 56 contigs were generated for three enterocolitica strains (Bat1, Bat2 and Bat3), with an average of 100–300x coverage across the genome. For comparative genomic analysis Trimmed reads were mapped to the reference genome sequence of the chromosome and the plasmid of *Y. enterocolitica* strain 8081 (GenBank accession numbers AM286415 and AM286416).

Contigs generated in CLC were processed using the EDGE bioinformatics platform version 2.3.1. For reference-based analysis RefSeq complete genomes *Y. enterocilitica* were select from EDGE drop-down list. The RAxML phylogenetic tree were constructed using EDGE pre-computed databases for Single Nucleotide Polymorphism (SNP) phylogeny analysis and two tree builders. Contigs were annotated using Prokka resulting in a total of 3092 genes (2884 coding genes). This Whole Genome Shotgun project has been deposited at DDBJ/ENA/GenBank under the accession JABMLT000000000, JABMLU000000000, and JABMLV000000000. The versions described in this paper are versions JABMLT010000000, JABMLU010000000, and JABMLV010000000. 

### 2.6. Phylogenetic Analysis

Phylogenetic and Molecular Evolution (PhaME) analysis tool v. 2.5.1 was used to generate a whole genome Single Nucleotide Polymorphism (SNP) alignment, based on genomic assemblies (contigs generated in CLC) and 77 genomes of *Y. enterocolitica* available from the public database PATRIC (Pathosystems Resource Integration Center). The complete genome of strain 8081 (Genbank accession number AM286415) was used as a reference. The whole-genome SNP alignments were then used to reconstruct a maximum likelihood-phylogenetic tree in RAxML (Randomized Axelerated Maximum Likelihood) program (v. 8.0.0) using 1000 bootstrap replicates using FastTree2.

The integrated software environment EnteroBase (http://enterobase.warnick.ac.uk) was used for identification of multi-locus sequence types (MLSTs) of *Y. enterocolitica* isolates obtained from bats in the context of worldwide *Y. enterocolitica* distribution. The trimmed sequence reads (FASTQ from CLC Bio) were uploaded onto the Enterobase under the *Yersinia* database. The sequence data were assembled and quality checked by the Enterobase’s pipeline. Size of assembly (number of contigs, total length and N50) were defined and then assembly was compared with other typing schemes presented in Enterobase, including wgMLST [12].

## 3. Results

### 3.1. Die-Off of Bats

A massive die-off of bats in the cave Tkibula-Dzevrula was reported by a local speleologist in April 2018. The same speleologist has also reported that three months earlier (January 2018), the colony of bats was evidently larger. Upon arrival to the cave, the field team found about 40 dead insectivorous bats and 150 live individuals. Dead animals were in different levels of decomposition and most of them were found in water. Less decomposed bats were selected and collected for the laboratory analysis. The collected 20 dead animals belonged to two species: the Schreiber’s bent-winged bat (*Miniopterus schreibersii*) and the greater horseshoe bat (*Rhinolophus ferrumequinum*). In February 2019, the field team visited this cave again. The activity of surviving bats at that time was low. Distantly, a number of bats in the cave were counted and the bats were identified as 35 lesser horseshoe bats (*Rhinolophus hipposideros*) and 15 greater horseshoe bats based on the visual recognition.

### 3.2. Identification of Bacteria in Dead Bats

The growth of bacterial colonies morphologically suspected as *Yersinia* was observed after 18–24 h incubation of three bat intestines. The characteristics of the colonies on CIN agar included a deep red center with transparent margins. Gram-stained microscopy showed Gram-negative bacillar-shaped microorganisms. The obtained isolates were cultured from small intestines of three of 11 tested Schreiber’s bent-winged bats (*M. schreibersii*). All three *Yersinia*-like isolates were identified as *Y. enterocolitica* based on the API29 test. None of the nine greater horseshoe bats tested positive using a *Yersinia*-culture test.

No growth of *Brucella* or *Francisella* bacteria was observed. Samples from bat carcasses contained many bacterial contaminants. Accompanying bacterial flora was present in all bat intestine samples and included *Bacillus cereus*, *Bacillus subtilis*, *Enterococcus faecalis*, *Hafnia alvei*, *Pseudomonas aeruginosa*, and *Serratia liquefaciens*.

### 3.3. Phenotypical Characterization of Y. entercolitica Strains Isolated from Bats

The three bat isolates were analyzed by API20 E. According to this test, two isolates (Bat1 and Bat2) had the index—1155723 (98.3% similarity), while the third isolate (BAT-3) had the API20E index 0055723 (99.4% similarity). The API tests results for three isolates obtained from Georgian bats are represented in Table 1.

### 3.4. Genome Sequences of Y. entercolitica Strains Isolated from Bats

The total assembly genome sizes of the *Y. enterocolitica* strains (Bat1, Bat2 and Bat3) were 4,675,988 bp (29 contigs); 4,675,119 bp (28 contigs); and 4,676,549 bp (56 contigs), respectively (Table 2). Reference-based analysis from the EDGE package drop-down list of 17 pre-selected reference genomes demonstrated that all the Bat strains were most similar to the *Y. enterocolitica* (type O:5) YE53/03 (NZ_NF571988_1) with an average nucleotide identity of 90% (Appendix A).

None of assembled contigs was aligned to the known *Y. enterocolitica* reference plasmids. A total of 4220–4225 predicted coding sequences (CDS) were detected, and about 70% of them were assigned to known functions. The analysis of virulence genes showed that all three Bat isolates contain *Yersinia* type 2 secretion 1 (*yts1*) (T2SS (Yst1)) cluster, chromosomally encoded type 3 secretion *Yersinia* secretion apparatus T3SS, virulence regulon transcriptional activator *virF*, *inv* invasin, *ystB*- *Yersinia* heat-stable enterotoxin type B, *yplA* phospholipase A, *yax*-cytotoxin YaxAB, direct heme uptake system protein genes—*hemP hemR*, *hmuV*, *hmuS*, *hmuT*, *hmuU*, flagella cluster I genes *flg*, protease phospholipase A *pla*, and fimbriae *myfA (psaA).* Missing classical virulence factors of *Y. enterocolitica* included genes for adhesin *ail*, insecticidal toxin complex-like protein *tccC*, virulence-associated vapC and VagC, and enterotoxin *ystA*. (Appendix A). 

No *Y. enterocolitica*-specific plasmids were detected from a gel electrophoresis conducted in all the bat strains according to Kado and Liu [11] for a plasmid screening (Appendix A). The absence of plasmids in the bat strains was also followed from the Next Generation Sequencing (NGS) data. None of the assembled contigs aligned to the known *Y. enterocolitica* plasmid references.

### 3.5. Phylogeny of Y. entercolitica Strains Isolated from Bats

The whole genome core sequences of the three strains *Y. enterocolitica* obtained during this study were compared with other genomes *Y. enterocolitica* available from a public database PATRIC using whole-genome SNP analysis (wg SNP) by PHAME with 77 reference strains. All three genomes from the bat strains (Bat1, Bat2 and Bat3) were grouped together according to the maximum likelihood phylogenetic SNP-based analysis. The Bat strains formed a distinct cluster within *Y. enterocolitica* Biotype 1A with the closest similarity to the environmental isolate IP2222 O:36/1A (water, Japan). *Y. enterocolitica* strains from the United Kingdom (UK) (YE13/03 ERR024563 O:6, 30), France (IP26014 ERR163895 1A O:7, 8, 19 Bovine), and Germany (YE53/30444 ERR024602 1A O:7, 8 Pig) as well as a human isolate from the UK (YE34/03 ERR024571 O:5) were clustered in distant proximity of the *Y. enterocolitica* bat isolates (Figure 3).

The EnteroBase platform containing hundreds of *Y. enterocolitica* strains was used to assess the MLST types of the Bat 1, 2, and 3 isolates with regard to *Y. enterocolitica* found worldwide. Both the whole genome multi locus sequence typing (wgMLST) and core genome multi locus sequence typing (cgMLST) demonstrated minor differences observed between the three bat genomes. All three genome sequences grouped together into the same cluster of the MStree/neighbor-joining either by SNPs or cgMLST analyses. In spite of increased resolution of the wgMLST compared to cgMLST, the former analysis also did not show any significant difference between the genomes (Appendix A).

According to the hierarchical clustering of cgMLST sequence types, the closest match to all three *Y. enterocolitica* strains from bats was a strain of *Y. enterocolitica* obtained from human fecal material in the UK (collection information includes year 2019; Bio Project ID_PRJNA481015; YER_DA2916AA_AS; ST3095sample ID_SAMN13283100; the secondary sample ID_SRS5645472; project ID_SRP154514). The 16S rRNA sequence of *Y. entercolitica* reported from a bat in Germany (GenBank under accession no. FN561632) could not be differentiated from the bat strains from Georgia. However, the small size of the ribosomal amplicon cannot prove identity with this strain.

## 4. Discussion

To the best of our knowledge, there has been only one report of *Y. entercolitica* in bats. Kristin Muhldorfer and her colleagues [8] tested 200 deceased bats of 16 species found in Germany during the 2006–2008 period. Of 25 bacterial genera cultured from bats, two cultures belonged to *Yersinia*, namely *Y. entercolitica* and *Y. pseudotuberculosis* [8]. The strain of *Y. entercolitica* (Y935) was isolated in pure culture from the spleen and intestine of a common pipistrelle (*Pipistrellus pipistrellus*).

Additionally, there were two more investigations, which culminated in the identification of *Y. pseudotuberculosis* in captive fruit bats that surprisingly belonged to the same species, Egyptian fruit bats (*Rousettus aegyptiacus*). Childs-Sanford et al. [13] described an outbreak of *Y. pseudotuberculosis* that occurred in a closed colony of Egyptian fruit bats at the Zoo in Syracuse, New York that resulted in the death of seven bats over a six-week period. Nakamura et al. [14] described an outbreak of yersiniosis in Egyptian fruit bats in the Zoological park in Japan caused by *Y. pseudotuberculosis* (serotype 4b) when 12 of 61 bats died.

Interestingly, all three referenced investigations reported isolation of *Y. entercolitica* or *Y. pseudotuberculosis* from dead bats, either collected in nature or from deceased captive bats. In two cases of investigations of captive bats, pathomorphological manifestations were noticed. Necropsy and histopathologic examination of a *Y. entercolitica* bat from Germany showed no inflammatory changes and the authors suggested a subclinical state of infection [8]. Tissues of dead bats collected in Georgia were too degraded for a special pathological evaluation. However, culturing of *Y. entercolitica* from the tissues of three bats provides grounds to suspect a causative role of this bacterium in the observed die-off and also to expect a higher infection when culturing from more fresh samples.

It is tempting to speculate that the Schreiber’s bent-winged bats found dead in the cave were more sensitive to the infection than horseshoe bats of the genus *Rhinolophus.* Such a speculation is based on the fact that all cultures were obtained from bats of this species. Interestingly, no surviving Schreiber’s bent-winged bats were found when the same cave was visited the following year. Nevertheless, the question of the etiological role of *Y. entercolitica* as a bat pathogen remains open until additional observations. Our observations raised the question about the impact of infectious bacterial agents on bat mortality that, according to Mühldorfer et al. [3], is largely unknown and has been neglected until now.

In Georgia, this was the first reported massive die-off of bats to our knowledge. It should be mentioned that the cave where the death of bats was recorded is located in an area of numerous massive karst caves. There are two known maternity colonies within eight kilometers from the Tkibula-Dzevrula Cave. One colony, consisting of about 300 individuals of Mediterranean horseshoe bat (*Rhinolophus euryale*), many Lesser horseshoe bat (*Rhinolophus hipposideros*) and single individuals of Lesser mouse-eared bat (*Myotis blythii*) and Schreiber’s bent-winged bat (*Miniopterus schreibersii*), is located in seven kilometers to the west of the Tkibula-Dzevrula cave, close to the village Tsutskvati, at the coordinates 42.271° N/42.852° E. Another maternity colony consisting of about 500 individuals of Lesser mouse-eared bat (*Myotis blythii*) is located 2.5 km south from the Tkibula-Dzevrula cave, at the coordinates 42.258° N/42.960° E. Consequently, the spread of any epidemic among the bats in this cave can be dangerous for colonies close by and generally for bat populations in western Georgia where karst massive is well developed with a large number of caves and known and unknown bat colonies.

The results of the phylogenic analysis indicate that the Georgian strains belonged to Biotype 1A. The strains of this biotype were once considered to not be pathogenic as they do not possess the virulence plasmid pYV and typically lack Ail, Myf, the Ysa T3SS, and Yst-a toxin [15,16]. The bat isolates from Georgia were also missing these virulence factors while containing several chromosomally located genes for T3SS apparatus highly homologous to those from other *Y. enterocolitica* strains. These Georgian isolates also possessed *inv* (i.e., invasion) and *ystB* (i.e., *Yersinia* heat-stable enterotoxin type B) genes, while lacking the attachment invasion locus *ail* gene, distinguishing these strains from the *ail*/*ystB* positive isolates described recently in the wild rodents in Poland [17]. An operon encoding fimbrial adhesin subunit MyfA (PsaA), usher and fimbria/pilus periplasmatic chaperon was also present in Georgian isolates. The MyfA was suggested to play an important role in promoting the adhesion to enterocytes, and was detected in some *Y. enterocolitica* biotype 1A strains isolated from clinical cases of yersiniosis and from patients with diarrhea [18]. Thus, there is evidence suggesting that *Y. enterocolitica* clinical isolates lacking classical virulence determinants, as well as some biotype 1A strains are virulent and can cause gastrointestinal diseases [15,19,20,21,22]. Additionally, Burnens et al. [23] reported that the virulence with biotype 1A *Y. enterocolitica* can be similar to those caused by pYV positive strains. Overall, infection caused by *Y. enterocolitica* biotype 1A can last for several weeks—even months—for all age groups, unlike strains with pYV plasmid, which are most common in children.

Though a role of the *Y. entercolitica* infection as the cause of the bats’ die-off has not been proven, the identification of bacteria in tissues of several dead bats can serve as indirect evidence suggesting this scenario. Clearly, more investigations including experimental studies are needed to prove this assumption. Further investigations of healthy populations of *M. schreibersii* can provide more information about distribution and prevalence of *Y. enterocolitica* in bats of this species. Meanwhile, the characterization of the novel strains cultured from bats can provide a clue for the determination of pathogenic properties specific for those strains.

## Figures and Tables

**Figure 1 microorganisms-08-01000-f001:**
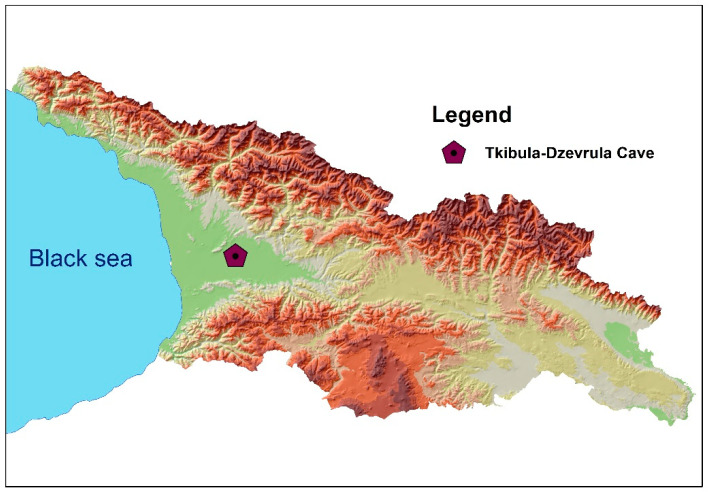
Location of the Tkibula-Dzevrula cave in western Georgia (Caucasus) where dead bats are found.

**Figure 2 microorganisms-08-01000-f002:**
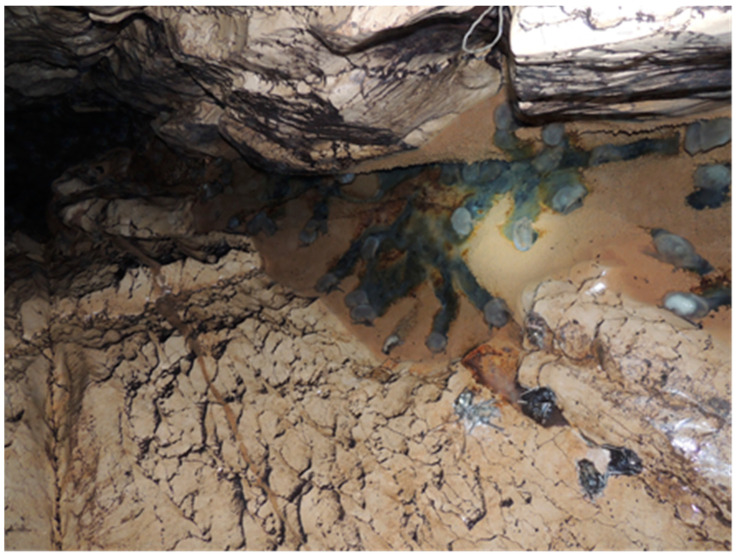
Picture of dead bats found in the Tkibula-Dzevrula cave.

**Figure 3 microorganisms-08-01000-f003:**
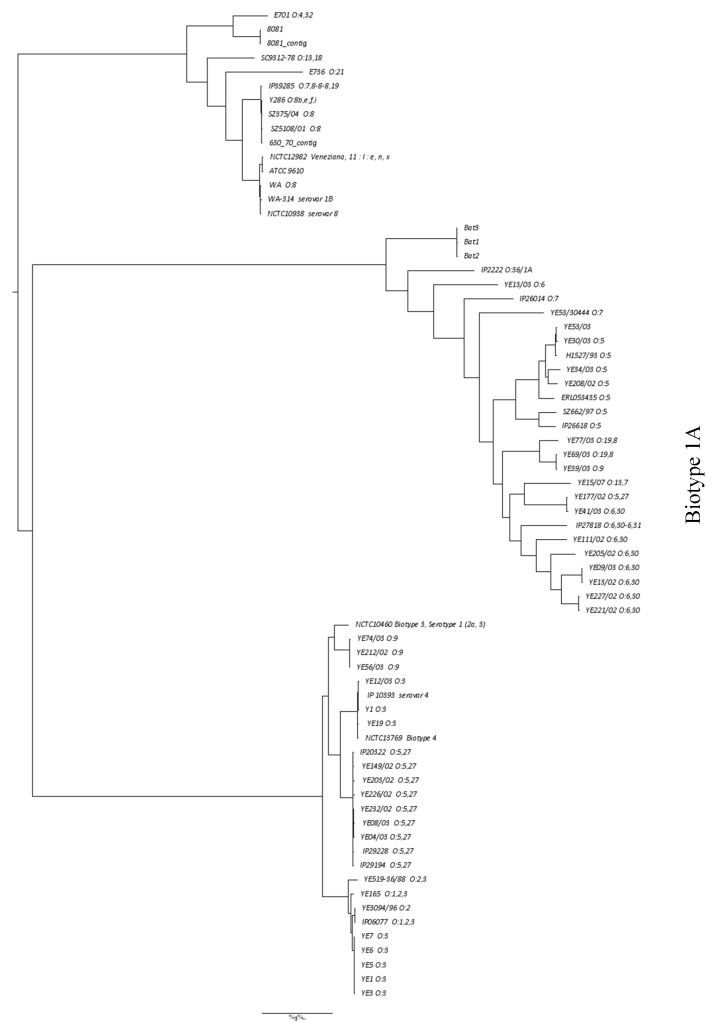
Phylogenetic relations between genomes of *Y. enterocolitica* strains (Bat1, Bat2 and Bat3) found in bats with previously reported 77 genomes from PATRIC database with whole-genome Single Nucleotide Polymorphism (SNP) analysis by Phylogenetic and Molecular Evolution (PHAME).

**Table 1 microorganisms-08-01000-t001:** Biochemical characterization of three strains of *Yersinia enterocolitica* from bats by the Analytic Profile Index (API) 20 E test.

Tests	Abbrev.	Bat 1	Bat 2	Bat 3
Test for beta-galactosidase enzyme	ONPG	pos	pos	neg
Decarboxylation by arginine dihydrolase	ADH	neg	neg	neg
Decarboxylation by lysine decarboxylase	LDC	neg	neg	neg
Decarboxylation by ornithine decarboxylase	ODC	pos	pos	neg
Utilization of citrate	CIT	neg	neg	neg
Production of hydrogen sulfide	H2S	neg	neg	neg
Urease test	URE	pos	pos	pos
Detection of tryptophan deaminase	TDA	neg	neg	neg
Indole test production	IND	pos	pos	pos
Voges-Proskauer test	VP	pos	pos	pos
Production of gelatinase	GEL	neg	neg	neg
Fermentation of glucose	GLU	pos	pos	pos
Fermentation of mannose	MAN	pos	pos	pos
Fermentation of inositol	INO	pos	pos	pos
Fermentation of sorbitol	SOR	pos	pos	pos
Fermentation of rhamnose	RHA	neg	neg	neg
Fermentation of sucrose	SAC	pos	pos	pos
Fermentation of melibiose	MEL	neg	neg	neg
Fermentation of amygdalin	AMY	pos	pos	pos
Fermentation of arabinose	ARA	pos	pos	pos

**Table 2 microorganisms-08-01000-t002:** Genome properties of *Yersinia enterocolitica* strains isolated from dead bats from Georgia.

Genomic Properties	Strain Bat1	Strain Bat2	Strain Bat3
Assembly size (bp)	4,675,988	4,675,119	4,676,549
* n* contigs	29	28	56
G + C content	47.35	47.09	47.09
Number of protein-coding genes (CDS)	4220	4220	4225
CDS (Function assigned)	2897	2897	2895
CDS (Hypothetical/putative)	1323	1323	1330
Genes	3107	3107	3105
Number of tRNAs	67	67	65
rRNAs (23 S, 16 S, and 5 S)	7	6	6

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
