# Peer review of "Identification of a Novel Yersinia enterocolitica Strain from Bats in Association with a Bat Die-Off That Occurred in Georgia (Caucasus)"

_microorganisms, 2020, doi:10.3390/microorganisms8071000_

Round 1
Reviewer 1 Report
This is an interesting paper that addresses a little studied area - pathogenic bacterial infections in wild bats. Further research on healthy M. schreibersii would be informative.
1 trivial typo; under "Funding" the word "Thread" should be "Threat".
Reviewer 2 Report
This is a well-written report on the occurrence of Y. enterocolitica in bats after a die-out in Georgia. The authors isolated bacteria from the gut of dead bats, identified Y. enterocolitica in some of the samples, and sequenced three isolates.
I have multiple smaller issues/items/questions:
- results line 161: if none of the greater horseshoe bats had the infection, is it then really legitimate to assume that there is a link between the (few) Y enterocolitica isolates and the observed die-off? I am not saying that this must be a wrong conclusion. But I am a bit disappointed that the "discussion" section does not pick up on this.
- results section 3.3 (phenotypic analysis): Please provide these results in a table. It is really difficult to read like this.
- Figure 3 would really benefit from an outgroup. I suggest that the authors add some Y pseudotuberculosis strains to the phylogeny?
- I am a bit unhappy with the results section where the presence/absence of known pathogenicity factors of Y enterocolitica is discussed (or not). What about sequence comparisons (of individual factors)? What about a table, where known factors are listed and their presence/absence in the bat strains is shown? Line 195 and around there has only a very brief (and disappointingly unexplained) list.
